

# Evaluation scheme design of college information construction based on a combined algorithm

Caiyou Shen[1], Yingjuan Shi[1] and Jing Fang[2]

[1] Institute of Cyberspace Security, Jinhua Advanced Research Institute, Jinhua, Zhejiang, China
[2] Department of Office, Jinhua Advanced Research Institut, Jinhua, Zhejiang, China

## ABSTRACT

By controlling the benefits and drawbacks of informatization construction (IC) and development, evaluating the level of education informatization (EI) development can aid in university administration and decision-making. This work develops an evaluation method for the University Information Construction (UIC) based on the Analytical Hierarchy Process (AHP) and the Particle Swarm Optimization-based back-Propagation Neural Network (PSO-BPNN) algorithm to address the fuzziness issue in grade evaluation in the IC. Firstly, a set of data-driven evaluation index systems of the UIC effect is constructed with 16 second-class indicators and four first-class indicators of infrastructure, resource management, information management, and safeguard measures. The AHP method is used to determine the weight of the first-class indicators of the IC model. Secondly, from two perspectives of inertia weight and learning factor, the PSO-BPNN algorithm is designed to fit and analyze the level of UIC. The experimental findings demonstrate that the proposed model's training impact is better, reflecting UIC's effectiveness more accurately.

## INTRODUCTION

To realize education fairness, increase education quality, and create a powerful nation with human resources, EI, which is the core meaning and fundamental characteristic of education modernization, is essential for China's educational reform and development. Colleges should be on the cutting edge of information management since they serve as the foundation for personnel training and technological innovation. The information management system should include teaching, scientific research, organization and management work, and colleges' future development planning. Based on big data and informatization, the quality of teaching and scientific research can be improved continuously, and achievements can be transformed into social productivity. Education and teaching, management, scientific research, and social services of the college constitute the key content of the development of basic EI. As the main realization space of EI, colleges provide scene support for the development of EI (*Guo, 2014*).

Additionally, the imbalanced issues caused by the disparity in educational information technology school growth may contribute to the disparities in regional, urban, and rural

Corresponding author
Jing Fang, jhgdyjy@163.com

area development. Therefore, under the background of EI 2.0, there is a need to build a systematic, scientific, and reasonable evaluation system for informatization (*Tang & Shen, 2020*).

There are many research results and cases on the scientific evaluation methods of UIC. The Summation of Tests for the Analysis of Risk (STAR) system in the United States and the self-assessment framework for school informatization (*Zhang, 2020*) and the guideline for ICT environment construction of schools in Heisei 30 years passed in Japan in 2018, which standardizes the evaluation of UIC (*Ministry of Education, Culture, Culture, Science and Technology , 2018*). In China, basic research is conducted on national and provincial research topics, primarily focusing on educational means, infrastructure, application level, decision analysis, and model architecture. A set of theoretical evaluation systems of higher EI in China has been preliminarily constructed. In addition, college evaluation methods pay more attention to the analysis and evaluation of the unilateral index system, especially the lack of application and research on the comprehensive index system of universities (*Liu, Cai & Yang, 2013*; *Phamn, Doa & Nguyenq, 2021*). At the same time, the algorithm construction mainly contains fuzzy mathematics, AHP, index aggregation, neural networks, and other methods. It is easy to be affected by the fuzziness of subjective judgment or grade evaluation. It is difficult to reflect the specific changes of IC, or it is computationally complex and difficult to realize. Therefore, this article designs the evaluation system of UIC based on the AHP and PSO-BPNN algorithm, aiming to provide new ideas for constructing EI.

## RELATED WORK

### Evaluation index

At present, many colleges adopt four stages of EI strategic planning goals. These include the primary construction goals, IC skills training of teachers and infrastructure, and promoting the construction of essential IC tools. The college innovation goals for IC programs, the strengthening goal of cultivating students' collaborative learning and autonomous learning ability through IC, and information literacy goals for teacher training and cooperative learning (*Sabiri, 2020*). To provide a more in-depth evaluation of teachers' education level from the three dimensions of informatization and sustainable education.

After the formulation of the 12th Five-Year Plan, China's overall education information infrastructure construction and development level have been greatly improved, but there is still a serious imbalance between regions. After a detailed analysis of different aspects of the framework (*Ministry of Education, 2013*), the National Education Statistics in 2013 added information statistical indicators such as "Access to the Internet," "Multimedia utilization rate," and "Network multimedia classroom" for teachers' teaching and students' learning. In 2019, the EI and network security worked to refine the requirements for college network and information technology applications (*Ministry of Education, 2019*).

The existing evaluation index system is mainly aimed at the overall informatization development of colleges. Still, it is only a dimension in the evaluation of college informatization. However, the relevant evaluation indicators of management

informatization are relatively simple, and there is no rich and comprehensive evaluation index system specifically for the EI. Most of the current indicators of EI are formulated for the result data, ignoring the potential value of process data. Since the big data era has arrived, we can fully utilize its potential and consider both the process and data when evaluating EI.

### EI evaluation

In the field of EI evaluation, researchers have been applying new statistical methods and introducing computer technology. Different evaluation models are used to realize the application of varying evaluation scenarios. *Lucas, Promentilla & Ubando (2017)* introduced AHP to quantify the effect of teacher EI training according to the preference value of participants. *Tao, Wei & Xu (2021)* used the combined algorithm to obtain the subjective and objective evaluation weights. He then adopted the improved augmented Lagrange multiplier algorithm to get the combination optimization weights for a more effective evaluation. *Li et al. (2019)* used the Delphi method and AHP to construct the higher EI development level evaluation index system and conducted empirical research at Shanghai University. *Wang (2008)* used the AHP and grey theory to comprehensively evaluate EI.

In recent years, many scholars have applied the neural network in education. *Lino, Rocha & Sizo (2019)* applied BPNN to an automatic evaluation in a virtual teaching environment, optimized it by Bayesian regularization, and simulated expert scoring through different evaluation models. *Zhike (2013)* uses the immune genetic algorithm to improve the standard BPNN algorithm. Taking English teaching performance evaluation data from three universities as examples, the effectiveness and feasibility of the improved algorithm are verified. *Huang (2013)* applied the improved BPNN to the evaluation of basic EI, completed the modeling and implementation of the effectiveness evaluation of EI, and proved that the application of BPNN in the performance evaluation of basic EI has high stability. *Peng (2012)* used BPNN to build a relevant mathematical model, applied it to identifying education quality evaluation grades in ethnic colleges, and realized a more scientific and reasonable evaluation mode.

# DESIGN OF EVALUATION SYSTEM FOR UIC

The IC model of colleges is built on the foundation of level optimization. The essential process characteristic qualities for each model level, including specific evaluation indicators, are listed below.

## Overall design

The specific factors affecting the system planning of the UIC model include infrastructure, resource construction, IC and safeguard measures. Various factors play a role in the initial stage, development stage and optimization stage of the model. Each level of the model contains several key process areas. The process domain is the boundary range of the feature attribute set. The IC model's current status can be identified by judging the internal process domain characteristics. The key process area is one of the essential bases for constructing the

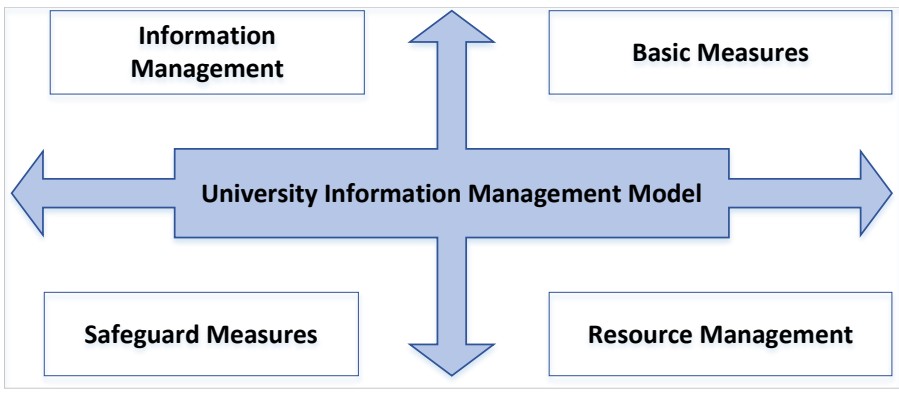

**Figure 1** The internal relationship of UIC.

model index evaluation. It can make the index evaluation system reasonable and complete and meet the dynamic requirements. The four-quadrant method is used to study the relationship between the factors affecting the construction of the UIC and the correlation between each factor of the whole model, as shown in Fig. 1.

The infrastructure factors in Fig. 2, such as the construction of the multimedia network and the investment of various hardware resources, play a fundamental role. The main focus of UIC is resource construction, which includes creating and updating multiple educational resources, research resources for science, and evaluation indicators.

## Architecture design

The evaluation system mainly includes a data acquisition, evaluation, and display layer. The system architecture is shown in Fig. 2.

The bottom layer of the system architecture is the data layer, which stores all kinds of data of the evaluation system, including evaluation index, index weight, process, result and fundamental data. The evaluation index data is to input the index system's name and each index's grade into the database. Index weight data refers to the proportion of each index in the overall weight. Process data refers to the data generated in the process of user operation. This part of the data eliminates the error caused by human subjectivity, which is the most important part of the data layer and the key data for information evaluation. Finally, result data refers to the data reflecting the IC, such as network bandwidth, consumable funds, etc., while the primary data include the user and organization table.

Data acquisition includes automatic and manual acquisition in the relevant layers. Automatic acquisition uses a docking interface to collect relevant evaluation data, which is easy to implement with open-source software such as Flume. In contrast, manual collection refers to data submitted by appropriate personnel and cannot be automatically acquired by the system, such as information disclosed on websites and in the media. Data cleaning is converting noisy data into normal data and preparing data for information evaluation.

The evaluation layer's primary function is data-based evaluation and calculation. According to the evaluation algorithm, the MapReduce program is compiled, and a big data cluster calculates the development of indicators. The display layer mainly uses

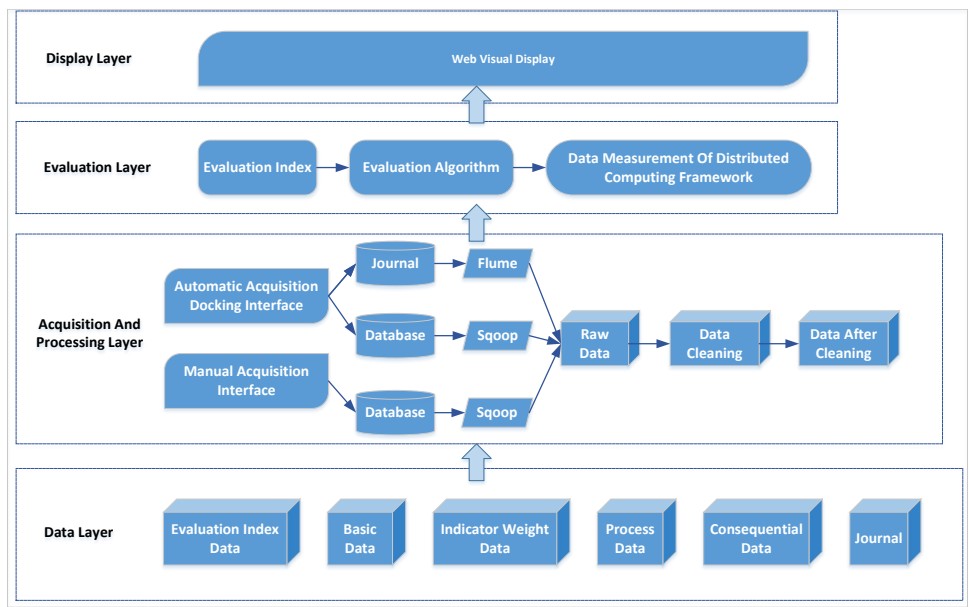

**Figure 2  System architecture.**

visualization technology to display the evaluation results of the system, including various statistical charts.

## Evaluation index

All evaluation index documents issued by the superior departments play a guiding and exemplary role in the UIC. Still, there are more or fewer constraints among each index system. To fully reflect the achievements of IC, we need to implement three aspects of work: (1) establishing the basic framework model, organizing the index system and relationship, clarifying the framework, and standardizing the process; (2) an optimal combination method is found between the constraints and the corresponding evaluation indexes to balance the weight relationship of each scoring system to obtain the optimal solution, which reflects the unity of objective evaluation and subjective experience.

To build a good evaluation system, it is necessary to determine the weight ratio of each index. Big data tools should be used to screen the indicators of the evaluation system, and the correlation between indicators and the intrinsic correlation between indicator elements should be intensely mined. This article establishes an index evaluation system based on infrastructure, resource management, information management, and safeguard measures, which takes the four factors as the first level evaluation index. Then it refines each level index into several secondary indicators. The evaluation index system of UIC is shown in Fig. 3.

As the social, economic and network environments are dynamic. The model's validity and dependability should be periodically assessed, along with the primary and secondary indicators and the weight proportion relationship between the index systems, to ensure that the information management model system is in step with the growing network trend.

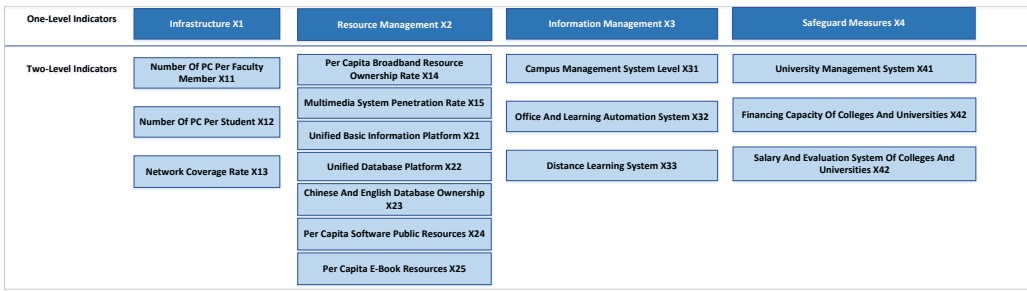

**Figure 3  Index system of UIC.**

## Evaluation scheme

The comprehensive evaluation module measures the collected and cleansed data, and the calculation results will reflect the informatization level of college education management. This study uses the comprehensive index method to evaluate the education management of UIC. Due to the weak correlation between the indicators, the difference between the index values is not big, and the importance of each index is relatively uniform. The linear weighted model can be used to calculate the comprehensive index, as shown in Eq. (1):

$$EDI = \sum_{i=1}^{n} W_i \left( \sum_{j=1}^{m} W_{ij} \left( \sum_{k=1}^{r} W_{ijk} Z_{ijk} \right) \right) \tag{1}$$

EDI represents the information level of college education management, which is directly proportional to the information level of UIC; $n$ is the number of first-level core indicators of the UIC level; $m$ is the number of $i^{th}$ class indicators of the UIC level; $r$ is the index number of $j^{th}$ class index of the UIC level, representing the number of the final index; $W_i$ is the weight of $i$th class index in the total index, and $\sum_{i=1}^{n} W_i = 1$; $W_{ij}$ represents the weight of the $j^{th}$ index in the $i^{th}$ index, and $\sum_{j=1}^{n} W_{ij} = 1$; $Z_{ijk}$ is the infinitely toughened value of the $k^{th}$ index of the $j^{th}$ index of the $i^{th}$ index, and $\sum_{k=1}^{r} W_{ijk} = 1$.

# INDEX EVALUATION MODEL BASED ON AHP AND PSO-BPNN ALGORITHM

## First-level index evaluation

Firstly, the AHP method is used to determine the weight set $\omega = \{\omega_1, \omega_2, \omega_3, \omega_4\}$ of the first-level indicators of the information construction model, where $\omega_1$ is the weight of infrastructure, $\omega_2$ is the weight of resource management, $\omega_3$ is the weight of information management and $\omega_4$ safeguard measures. The Hadoop tool is used to study the original data of colleges related to the primary index to determine the importance weight. Four factors form the importance matrix of index weight

$$H = \left[ h_{ij} \right] = [h_{11} h_{12} h_{21} h_{22}] \tag{2}$$

where $h_{11}$ is information management, $h_{12}$ is infrastructure, $h_{21}$ is safeguard measures, $h_{22}$ is resource management.

Then determine the importance of each secondary evaluation index under the first-level evaluation index. Through the weight relationship between the first-level index and the second-level index, and calculate the maximum characteristic root $\lambda$ of matrix $H$, then the consistency index $\eta$ of UIC model can be expressed as:

$$\eta = \frac{\lambda_{max} - n}{m - 1} \tag{3}$$

where $m$ is the number of matrix state vectors, $n$ is the number of evaluation objects. When the consistency index value $\eta < 0.1$, it is considered that the design of the first-level index of the UIC evaluation system is reasonable.

## Secondary index evaluation
### *Algorithm design*
To improve the ability and stability of the BPNN model, the PSO algorithm is combined with the BPNN. The performance of the BPNN model is greatly affected by each node's connection weight and the threshold value. In the actual training and application process of the evaluation model of UIC level based on BPNN, the generation of the initial threshold and connection weight of the BPNN model is random. Through the PSO-BPNN model, the position of each particle is used to replace the weight and threshold of each node of the neural network until the particles are constantly searching, the optimal parameter solution with the minimum training error of the neural network model is calculated, to improve the training effect, solution speed and generalization ability of BPNN. The formal description is as follows:

In a given D-dimensional space, set the number of particles $M$, then in a particle species group $T = \{x_1, x_2, \ldots, x_m\}$, the velocity of the $i$ th particle in D-dimensional space is represented as $V_i = \{v_{i1}, v_{i2}, \ldots, v_{iD}\}$, the position is represented by $X_i = \{x_{i1}, x_{i2}, \ldots, x_{iD}\}$. The initial velocity of a particle determines the direction and speed of its motion, and the position reflects the position of the solution represented by the particle in the solution space, which is the basis for evaluating the mass. The best position found by the $i$ th particle is denoted by $P_i = \{P_{i1}, P_{i2}, \ldots, P_{iD}\}$, and the best position found by all particles in the particle population is denoted by $P_g = \{P_{g1}, P_{g2}, \ldots, P_{gD}\}$, The particle $x_i$ updates its velocity and position in each dimension according to Eqs. (4) and (5).

$$v_{ij}^{t+1} = \omega v_{ij}^t + c_1 r_1 \left( p_{ij}^t - x_{ij}^t \right) + c_2 r_2 \left( p_{gj}^t - x_{ij}^t \right) \tag{4}$$

$$x_{ij}^{t+1} = x_{ij}^t + v_{ij}^{t+1} \tag{5}$$

where $j = 1, 2, \ldots, D, i = 1, 2, \ldots, Mi$, $v_{ij}^{t+1}$ is the velocity of the $i$ th particle in the $j$ th dimension at the $t$ th iteration. $x_{ij}^t$ is the position of the $i$ th particle in the $j$ th dimension at the $t$ th iteration; $\omega$ is the inertia weight, $c_1$ and $c_2$ are the learning factors; $r_1$ and $r_2$ are independent random numbers between [0,1]. In order to prevent $x_{ij}^t$ out of scope, the need to control the scope of $v_{ij}^{t+1}$, when $|V_i(t)| < V_{max}$, it values for $V_i(t)$, when $|V_i(t)| \geqslant V_{max}$, it values for $V_{max}$.

### *Optimization process*
An appropriate value of $w$ can improve the algorithm's performance, the ability to find the optimal solution, and the reduction of the number of iterations. Inertia weight adjustment

strategies mainly include linear decreasing and increasing strategies, nonlinear reducing and adaptive adjustment strategies, etc. This article adopts the adaptive dynamic adjustment strategy to change the inertia weight. The formula is as follows:

$$w = w_{\min} - \frac{(w_{\max} - w_{\min}) * (f_i - f_{\min})}{f_{a_g} \cdot f_{\min}}, f_i \leq f_{\omega_g} \tag{6}$$

$$w = w_{\max}, f_i > f_{a_g} \tag{7}$$

where $w_{\max}$ is the maximum value of inertia weight, $w_{\min}$ isthe minimum value of inertia weight, $f_i$ is the particle adaptation value, the optimal particle adaptation value is $f_m$, the average particle swarm adaptation value $f_{\omega_g} = \frac{1}{n}\sum_{i=1}^{n}f_i$.

In the learning factor, the value range of $C_1$ and $C_2$ is [0,4]. The fixed value will generally cause premature local extreme value. Therefore, dynamic adjustment of $C_1$ and $C_2$ should be adopted, and the formula is as follows:

$$C_1 = 2\sin^2\left[\frac{\pi}{2}\left(1 - \frac{t}{T_{max}}\right)\right] \tag{8}$$

$$C_2 = 2\sin^2\left(\frac{\pi t}{2T_{max}}\right) \tag{9}$$

where the number of iterations is t, and $T_{max}$ is the maximum number of iterations of the particle swarm.

## EXPERIMENT AND ANALYSIS

### System performance test

Taking five universities in a city with an established model as research samples (statistics time is from October 2018 to September 2021), the function and performance of each module of the model are verified. This article mainly tests the system performance from two aspects: the response time of the system and the number of users simultaneously. For a single program, the duration of entering the system and retrieving information is less than 5s to ensure a better user experience. The response time of 10 clients was counted, and the traditional evaluation model was introduced for comparison. The change curve of time response is shown in Fig. 4.

Figure 4 shows that it takes 2s to open a single program, 9s to open five programs, and 13s to open ten programs; While in the traditional static mode, they were 8s, 19s, and 23s, respectively. Compared to the other models, the efficiency of the proposed model is higher. In addition, the capacity of the IC evaluation system based on a combined algorithm is more robust.

### Model validation

#### Parameter setting

The parameters of the PSO algorithm mainly include: population size N having a value of 20; inertia weight $\omega$, generally 0.9; learning factors C1 and C2, value 2.0; r1 and r2 are

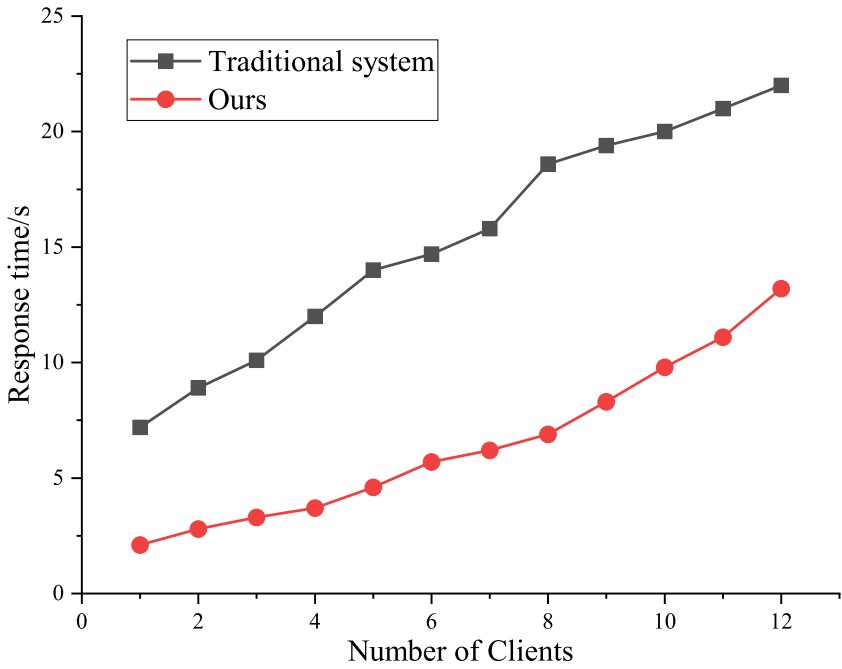

**Figure 4** Results of system test.

random numbers in the interval [0,1]; the value of the maximum particle velocity $V_{max}$ is 20; and the error accuracy is 0.001.

Iris data set was used to verify the model. The BPNN structure was designed to set four neurons in the input layer, five in the hidden layer, and three in the output layer. The excitation function of the hidden layer is Sigmoid, and the error rate is 0.001.

### Results and discussion

The convergence of the PSO-BPNN model is shown in Fig. 5. The epochs of the BP algorithm in the iris dataset is 69 that of the standard PSO-BP algorithm is 59. Whereas it is 50 for the improved PSO-BPNN algorithm, which can be seen that the convergence speed of the model proposed in this article is faster when the convergence objectives are the same. The prediction effect of the model is shown in Fig. 6.

It can be seen that the improved PSO-BPNN algorithm has a better effect than BP and PSO-BP algorithms on the prediction output. In addition, the evaluation model of UIC based on the PSO-BPNN has a good result, and its fitting degree and trend are consistent with the expected value. This indicates that the generalization ability of the PSO-BPNN model is higher than that of the original BPNN model based on subjective and objective comprehensive evaluation data. When the amount of data is larger, the PSO-BPNN can be used to ensure the stability and accuracy of the evaluation. Still, it can also improve evaluation efficiency, which provides a new idea for evaluating the development level of university informatization.

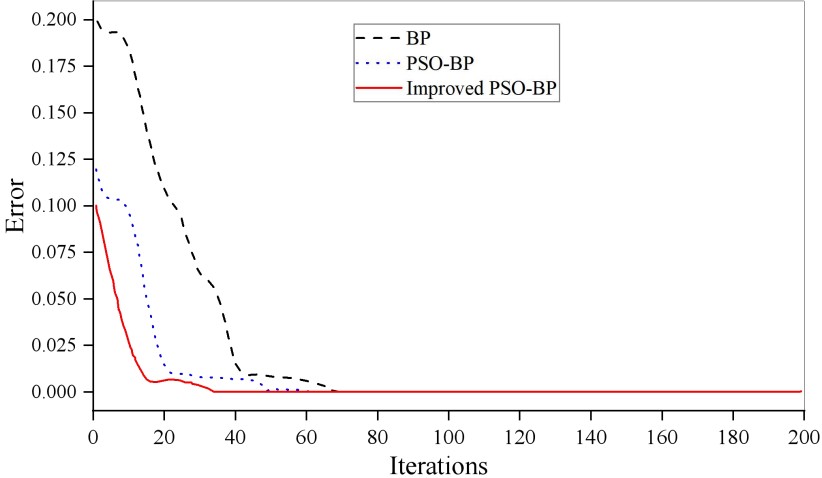

**Figure 5** Model training effect.

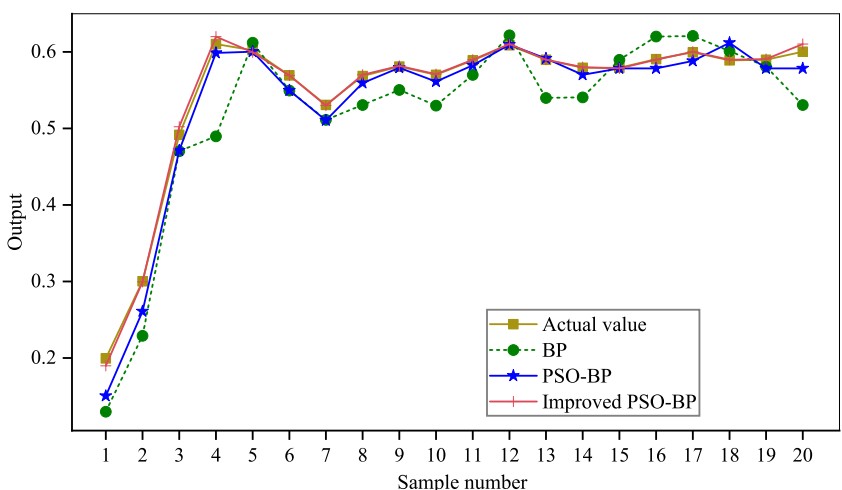

**Figure 6** Prediction effect of the model.

## Evaluation effect of UIC

Based on the first-level index weights of big data evaluation samples of a college in the past four years, this article evaluates the four aspects of infrastructure, resource management, information management, and safeguard measures. The results are shown in Fig. 7.

According to the comprehensive score obtained, combined with the known experience and achievements of UIC, it is clear that the investment in facility construction in the past four years has improved the nominal index to a certain extent but combined with the influence of other related factors, the total score has not achieved synchronous growth. In terms of information support, the score has been continuously improved due to the improvement of the operating environment and the development of curriculum

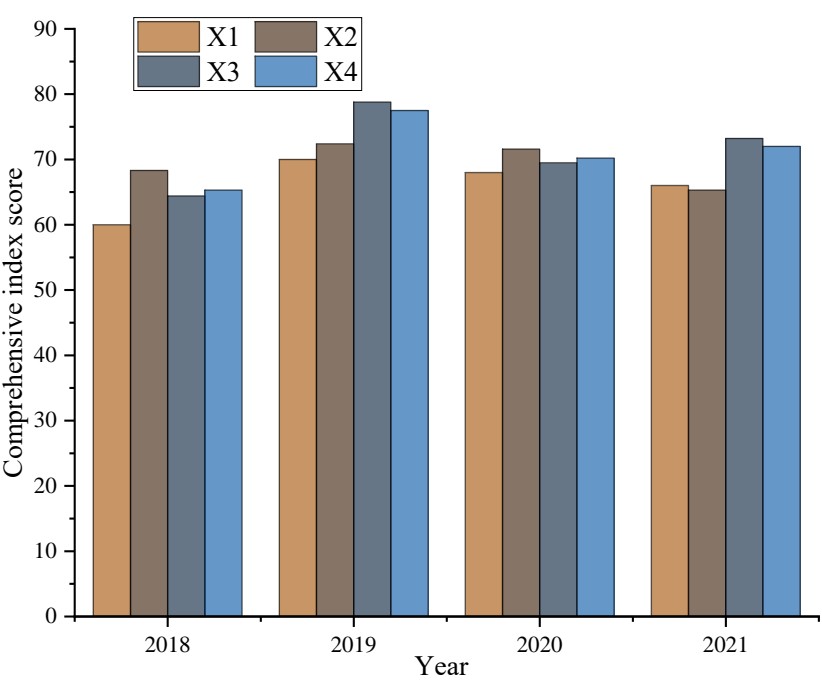

**Figure 7** Evaluation effect of UIC.

construction. Still, its growth trend has slowed with the extension of time, which means that if we want to ensure the effect of IC, we must consider making long-term continuous adjustments or investments in the system, which is the focus of the future UIC. In addition, regarding information application and data resources, the overall improvement is not evident due to the weight control, which reflects the significance of the combined weight algorithm. Because of the adjustment of the combined weight of the support and maintenance system, the work and evaluation results show synchronous correlation changes, which further verifies that the evaluation model can effectively reflect the effectiveness of UIC.

## CONCLUSION

This article focuses on the evaluation model of UIC based on the AHP and PSO-BPNN algorithm and realizes the nonlinear evaluation of multidimensional indexes. The evaluation system of UIC based on a combined algorithm reflects the achievements and connotations of UIC. The experimental results show that the evaluation model can effectively reflect the effectiveness of UIC. The IC evaluation system based on a combined algorithm has a more robust capacity, which can be applied and promoted as an effective tool of a university evaluation system. In future research, we will optimize the current evaluation indicators and take more examples for verification.

### Funding
The authors received no funding for this work.

### Competing Interests
The authors declare there are no competing interests.

### Author Contributions
- Caiyou Shen conceived and designed the experiments, performed the experiments, analyzed the data, performed the computation work, prepared figures and/or tables, authored or reviewed drafts of the article, and approved the final draft.
- Yingjuan Shi conceived and designed the experiments, performed the experiments, analyzed the data, performed the computation work, prepared figures and/or tables, authored or reviewed drafts of the article, and approved the final draft.
- Jing Fang conceived and designed the experiments, performed the experiments, analyzed the data, performed the computation work, prepared figures and/or tables, authored or reviewed drafts of the article, and approved the final draft.

### Data Availability
The code is available in the Supplemental Files.

The five university information datasets are available at Kaggle:

- https://www.kaggle.com/datasets/Cornell-University/arxiv
- https://www.kaggle.com/datasets/rocki37/open-university-learning-analytics-dataset
- https://www.kaggle.com/datasets/rtatman/the-national-university-of-singapore-sms-corpus
- https://www.kaggle.com/datasets/anlgrbz/student-demographics-online-education-dataoulad
- https://www.kaggle.com/datasets/mohaiminul101/international-students-in-china

### Supplemental Information
Supplemental information for this article can be found online at http://dx.doi.org/10.7717/peerj-cs.1327#supplemental-information.

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
