# Peer review of "Evaluation scheme design of college information construction based on a combined algorithm"

_PeerJ Computer Science, doi:10.7717/peerj-cs.1327_

## Round 0.1 · original submission · Major Revisions

Dear Authors,

Thanks for your good work submission, the experts are giving some major concerns to be addressed. Therefore, please carefully revise the paper and re-submit after incorporating all the comments. Thank you

Reviewer 1 ·

Basic reporting

This paper designs the evaluation system of university information construction based on AHP and PSO-BP algorithms. Firstly, a set of data-driven evaluation index systems of university informatization construction effect is constructed. From two perspectives of inertia weight and learning factor, the PSO-BP neural network algorithm is designed to fit and analyze the level of university information construction. The training effect of the evaluation model is better, which can effectively reflect the effect of university information construction and provide a new idea for the construction of education information. To be successfully accepted by this journal, the article still needs to be improved in the following aspects
A. The keywords of the article are not representative enough. The author should further extract the keywords in line with the topic of the article according to the research content
B. The language expression of the conclusion part needs to be optimized, and the content of this part needs to supplement the limitations and future development direction.
C. Much of the introduction focuses on the meaning of Informatization construction of universities in other countries. In addition, it is necessary to clearly state the related content of China and briefly describe the experiments, techniques and methods applied to the research-related problems
D. In the description of the system structure, there is a lack of analysis and introduction of the data layer, and the corresponding explanation of process data/result data is lacking
E. It is suggested that the authors delete the simple expression in the data analysis (Line 319-327) and pay more attention to the variation trend between the data and its specific meaning. At present, the depth of the analysis is insufficient
F. What is the key to evaluating the problem solved in the process of algorithm design? This is the key to the combination of the PSO algorithm and BP neural network
G. The English of your manuscript must be improved before resubmission. We strongly suggest that you obtain assistance from a colleague who is well-versed in English or whose native language is English.

Experimental design

This paper designs the evaluation system of university information construction based on AHP and PSO-BP algorithms. Firstly, a set of data-driven evaluation index systems of university informatization construction effect is constructed. From two perspectives of inertia weight and learning factor, the PSO-BP neural network algorithm is designed to fit and analyze the level of university information construction. The training effect of the evaluation model is better, which can effectively reflect the effect of university information construction and provide a new idea for the construction of education information. To be successfully accepted by this journal, the article still needs to be improved in the following aspects
A. The keywords of the article are not representative enough. The author should further extract the keywords in line with the topic of the article according to the research content
B. The language expression of the conclusion part needs to be optimized, and the content of this part needs to supplement the limitations and future development direction.
C. Much of the introduction focuses on the meaning of Informatization construction of universities in other countries. In addition, it is necessary to clearly state the related content of China and briefly describe the experiments, techniques and methods applied to the research-related problems
D. In the description of the system structure, there is a lack of analysis and introduction of the data layer, and the corresponding explanation of process data/result data is lacking
E. It is suggested that the authors delete the simple expression in the data analysis (Line 319-327) and pay more attention to the variation trend between the data and its specific meaning. At present, the depth of the analysis is insufficient
F. What is the key to evaluating the problem solved in the process of algorithm design? This is the key to the combination of the PSO algorithm and BP neural network
G. The English of your manuscript must be improved before resubmission. We strongly suggest that you obtain assistance from a colleague who is well-versed in English or whose native language is English.

Validity of the findings

This paper designs the evaluation system of university information construction based on AHP and PSO-BP algorithms. Firstly, a set of data-driven evaluation index systems of university informatization construction effect is constructed. From two perspectives of inertia weight and learning factor, the PSO-BP neural network algorithm is designed to fit and analyze the level of university information construction. The training effect of the evaluation model is better, which can effectively reflect the effect of university information construction and provide a new idea for the construction of education information. To be successfully accepted by this journal, the article still needs to be improved in the following aspects
A. The keywords of the article are not representative enough. The author should further extract the keywords in line with the topic of the article according to the research content
B. The language expression of the conclusion part needs to be optimized, and the content of this part needs to supplement the limitations and future development direction.
C. Much of the introduction focuses on the meaning of Informatization construction of universities in other countries. In addition, it is necessary to clearly state the related content of China and briefly describe the experiments, techniques and methods applied to the research-related problems
D. In the description of the system structure, there is a lack of analysis and introduction of the data layer, and the corresponding explanation of process data/result data is lacking
E. It is suggested that the authors delete the simple expression in the data analysis (Line 319-327) and pay more attention to the variation trend between the data and its specific meaning. At present, the depth of the analysis is insufficient
F. What is the key to evaluating the problem solved in the process of algorithm design? This is the key to the combination of the PSO algorithm and BP neural network
G. The English of your manuscript must be improved before resubmission. We strongly suggest that you obtain assistance from a colleague who is well-versed in English or whose native language is English.

Additional comments

This paper designs the evaluation system of university information construction based on AHP and PSO-BP algorithms. Firstly, a set of data-driven evaluation index systems of university informatization construction effect is constructed. From two perspectives of inertia weight and learning factor, the PSO-BP neural network algorithm is designed to fit and analyze the level of university information construction. The training effect of the evaluation model is better, which can effectively reflect the effect of university information construction and provide a new idea for the construction of education information. To be successfully accepted by this journal, the article still needs to be improved in the following aspects
A. The keywords of the article are not representative enough. The author should further extract the keywords in line with the topic of the article according to the research content
B. The language expression of the conclusion part needs to be optimized, and the content of this part needs to supplement the limitations and future development direction.
C. Much of the introduction focuses on the meaning of Informatization construction of universities in other countries. In addition, it is necessary to clearly state the related content of China and briefly describe the experiments, techniques and methods applied to the research-related problems
D. In the description of the system structure, there is a lack of analysis and introduction of the data layer, and the corresponding explanation of process data/result data is lacking
E. It is suggested that the authors delete the simple expression in the data analysis (Line 319-327) and pay more attention to the variation trend between the data and its specific meaning. At present, the depth of the analysis is insufficient
F. What is the key to evaluating the problem solved in the process of algorithm design? This is the key to the combination of the PSO algorithm and BP neural network
G. The English of your manuscript must be improved before resubmission. We strongly suggest that you obtain assistance from a colleague who is well-versed in English or whose native language is English.

Reviewer 2 ·

Basic reporting

This research realizes the nonlinear evaluation of multidimensional indexes. The evaluation system of university information construction based on a combination algorithm reflects the achievements and connotations of university information construction. The experimental results verify that the evaluation model can effectively reflect the effect of university information construction, and the information construction evaluation system based on the combination algorithm has a stronger accommodating capacity, can guide the construction of smart universities, and can be applied and promoted as an effective tool of the university evaluation system. However, there are also some problems, some revisions need to be revised to make sure that the manuscript can be accepted. The common problems are as follows:
1. There are some problems in the language expressed in this paper, which needs to be modified. Please check the Chinese characters in the replacement formula and the redundant space characters in the references.
2. The abstract of the article needs to be strengthened to be concise and comprehensive.

Experimental design

1. The author should carefully check the notes and footnotes of the formulas used in the paper, for example, Formula (2) all lack corresponding explanations.
2. The author's description of Figure 2 (system architecture) is not comprehensive enough, bringing readers great trouble.
3. What are the advantages of combining the PSO algorithm with the BP neural network? These outstanding contributions should be noted in the text

Validity of the findings

1. The conclusion part needs to adjust the language expression, and the elaboration of this part is too lengthy. The author needs to simplify this part.
2. Section 5.2.1 (Parameter setting) only introduces the training parameters of the PSO algorithm, but lacks the parameter setting of the BP algorithm
3. The author should give specific results relevant to the purpose; Avoid outcomes that are irrelevant to the purpose; Avoid vague expressions such as "relatively large" or "significantly different"

---

## Round 0.2 · Minor Revisions

Thank you for revising the paper according to the reviewers' comments. Your paper is scientifically suitable but still needs some language corrections, especially the abstract. Therefore, you are requested to revise the language of the article so that the readers can easily understand.

Reviewer 1 ·

Basic reporting

accepted

Experimental design

accepted

Validity of the findings

accepted

Reviewer 2 ·

Basic reporting

The revision has addressed my concerns.

Experimental design

The revision has addressed my concerns.

Validity of the findings

The revision has addressed my concerns.

---

## Round 0.3 · Minor Revisions

Thank you for improving the paper according to the comments. Good luck for your future research.

---

## Round 0.4 · accepted · Accept

Thank you for your revision, your paper is acceptable for publication. Recommend a final proofread to catch the *very few* remaining typos.